# Re-thinking "non-response" to wasting treatment: Exploratory analysis from 14 studies

Cécile Cazes[1,2], Heather Stobaugh[3,4], Paluku Bahwere[5], Paul Binns[6], Robert E. Black[7], Erin Boyd[4,8], André Briend[9,10], Sheila Isanaka[11], Suvi T. Kangas[12], Tanya Khara[1], Natasha Lelijveld[1], Martha Mwangome[13,14], Mark Myatt[15], Gloria Odei Obeng-Amoako[16], Indi Trehan[17], Philip T. James[1]*

1 Emergency Nutrition Network, Oxford, United Kingdom, 2 University of Bordeaux, National Institute for Health and Medical Research, Research Institute for Sustainable Development, Bordeaux Population Health Research Centre, Bordeaux, France, 3 Action Against Hunger USA, New York City, New York, United States of America, 4 Friedman School of Nutrition Science and Policy at Tufts University, Boston, Massachusetts, United States of America, 5 Center for Epidémiology, Biostatistics and Clinical Research, School of Public Health, Université Libre de Bruxelles, Brussels, Belgium, 6 Action Against Hunger UK, London, United Kingdom, 7 Johns Hopkins Bloomberg School of Public Health, Institute for International Programs, Baltimore, Maryland, United States of America, 8 U.S. Agency for International Development (USAID), Bureau for Global Health, Washington, DC, United States of America, 9 Department of Nutrition, Exercise and Sports, Faculty of Science, University of Copenhagen, Frederiksberg, Denmark, 10 Center for Child, Adolescent and Maternal Health Research, Faculty of Medicine and Health Technology, Tampere University and Tampere University Hospital, Tampere, Finland, 11 Harvard T.H. Chan School of Public Health, Boston, Massachusetts, United States of America, 12 International Rescue Committee, New York City, New York, United States of America, 13 Kenya Medical Research Institute, Centre for Geographic Medicine Research-Coast, Kilifi, Kenya, 14 Kenya Medical Research Institute, Wellcome Trust Research Programme, Kilifi, Kenya, 15 Brixton Health, Llwyngwril, Gwynedd, Wales, United Kingdom, 16 Department of Nutrition and Food Science, School of Biological Sciences, College of Basic and Applied Sciences, University of Ghana, Legon, Ghana, 17 Departments of Paediatrics, Global Health, and Epidemiology, University of Washington, Seattle, Washington, United States of America

* philip@ennonline.net

## Abstract

Children who receive therapeutic feeding for wasting treatment but do not reach the anthropometric definitions of recovery (usually within 12–16 weeks) are categorised as 'non-responders' and considered as treatment failures. We conducted a pooled analysis to explore the growth trajectories of non-responders and the appropriateness of the definition of 'non-response'. We pooled 14 studies of children aged 6–59 months receiving treatment for wasting. We included children classified by their studies as recovered or as non-responders. Observing the pooled data of non-responders' mid-upper arm circumference (MUAC), weight, weight-for-age z-score, weight-for-height z-score and daily weight gain rate, we found that the first quartile differentiated those who did not grow at all versus those that demonstrated some growth. We therefore defined 'low growth non-responders' as < 25th percentile anthropometric gain between admission and exit using the non-responders' pooled study data, and 'high growth non-responders' as ≥ 25th percentile gain. We plotted the growth trajectories of MUAC-, weight- and height-related indices of the recovered, high growth and low growth non-responder groups over time using mixed effects generalised additive models. We compared age, sex and anthropometric characteristics of the three groups and explored predictors of non-response category using a multivariate multinomial logistic

**Data availability statement:** This consolidated dataset comprises 14 studies. We obtained permission from each study principal investigator to use their data for purposes of this analysis, which aligned with original study consent. We do not have permission to make the consolidated dataset publicly available in a repository from either the study PIs or, more importantly, from the subjects in the original studies. Unfortunately this lies outside the ethical clearance we have, which is only to hold the joint dataset for the purpose of this specific analysis. However, each of the 14 studies has a published paper or protocol, detailed in our supplementary information and references, so that readers will be able to approach the original study principal investigators and request permission for the data from each included study. The organization imposing these ethical restrictions on the pooled dataset, on behalf of the study principal investigators, is the Emergency Nutrition Network. Further inquires on access to the minimal data set can be made by email to office@ennonline.net.

**Funding:** This paper is made possible by the generous support of the American people through the United States Agency for International Development (USAID) Bureau for Humanitarian Affairs (grant 720BHA23CA00001 to TK) and the Department of Foreign Affairs of Ireland (grant HQPCR/2024/ENN to TK). The contents are the responsibility of the authors and do not necessarily reflect the views of USAID, the United States Government or the Government of Ireland. The funders had no role in study design, data collection and analysis, decision to publish, or preparation of the manuscript.

**Competing interests:** I have read the journal's policy and the authors of this manuscript have the following competing interests: IT currently serves on the editorial board of PLOS Global Public Health. All other co-authors declare no competing interests.

regression model. For all outcomes, the high growth non-responders started with a worse anthropometric status compared to those who recovered, but then tracked along a near-parallel growth trajectory. The low growth non-responders showed limited growth throughout treatment. High growth non-responders are better viewed as 'delayed responders' and may need to be kept longer under treatment to recover and reduce the risks from early discharge. Low growth non-responders are the true treatment failures and should be referred for further investigations as quickly as possible. In conclusion, non-responders are not a homogenous group; ~75% of them respond well to treatment and ~25% are treatment failures.

## Background

Globally in 2022, acute malnutrition affected 45 million children aged under 5 years worldwide, including 14 million with severe wasting [1]. These figures are likely to be underestimates [2]. Undernutrition is an underlying cause of nearly half of all deaths among children under 5 years of age [3] and a leading risk factor for disability-adjusted life-years lost in low- and middle- income countries [4].

Mid-upper arm circumference (MUAC) and weight-for-height z-score (WHZ) are two indirect and imperfect indicators used as proxies to assess a child's overall nutritional status, categorise the severity of wasting, admit patients into acute malnutrition management programmes, and discharge them from treatment. Low MUAC and WHZ are both predictive of increased mortality risk [5] and retain great practical utility in operational programmes. Most community-based management of acute malnutrition (CMAM) protocols for treatment of wasting require children to reach MUAC ≥ 125mm, and/or weight-for-height z-score (WHZ) ≥ -2, generally for two consecutive visits, to be categorised as 'recovered' before discharge from therapeutic feeding. In the recently updated World Health Organization (WHO) 2023 guidelines [6], the recommended discharge criteria for children aged 6-59 months with severe wasting and/or nutritional oedema is reaching both WHZ ≥ -2 and MUAC ≥ 125mm, with an absence of nutritional oedema, for two consecutive weeks. The evidence to define discharge criteria using anthropometric indices has historically focused on the association of low MUAC with increased mortality for children, alongside the expected normal distribution of anthropometric indices at the population level.

Children who receive therapeutic feeding but do not reach the anthropometric definitions of recovery within a specified timeframe are generally categorised as 'non-responders' and considered as treatment failures. Many CMAM programmes use a pragmatic maximum length of stay in the programme that varies by context, but typically ranges from 12-16 weeks; some children are therefore discharged without reaching definitions of recovery. There has been some investigation into the predictors of non-response [7,8], but to our knowledge, a robust enquiry into the definition and growth trajectories of non-response using a global representation of treatment programmes has not been conducted previously.

Within CMAM programmes, we hypothesise that children classified as 'non-responders' are not homogeneous and may have different patterns of growth trajectories. Some may indeed not be growing at all, some may be slow to respond to treatment and need more time to recover, and others may be responding well and plateau in their growth just shy of the 'recovery' cut-off because they have reached their growth potential.

We conducted a pooled analysis of individual patient data to interrogate the appropriateness of existing definitions of 'non-response' for children receiving therapeutic feeding for wasting treatment. Our three main objectives were:

i)  to describe the patterns of growth among children classified as non-responders compared to those classified as recovered;

ii) to evaluate whether the programmatic maximum length of treatment (typically 12-16 weeks) is appropriate for categorising recovery and non-response; and,

iii) to describe the characteristics of non-responders at admission and assess their utility in predicting which children respond to treatment.

## Methods

### Study design

We conducted secondary analyses using a pooled dataset of 14 studies.

### Participants/ studies

The pooled dataset comprised 14 individual studies that contained data on children who received treatment for wasting. Eleven of these [9–19] were originally included in a previous analysis by the Wasting and Stunting Technical Interest Group (WaSt TIG) [20]. We added data from three additional studies [21–24]. S1 Table provides information on the 14 included studies, detailing the country, study design, intervention protocol, admission and exit criteria, and maximum length of stay in study, alongside the published reference for further contextual information.

 Datasets were eligible for inclusion if they met the following criteria:

i) Data were collected as part of a research study. Routine programme data were excluded.

ii) Provided individual child-level data for children aged 6–59 months admitted for outpatient management of uncomplicated moderate or severe wasting.

iii) Were described within a peer-reviewed article containing details of the study setting, programme implementation, data collection and original ethical approval.

iv) Included a minimum set of variables (child age, weight, height/length, MUAC and oedema status at admission and follow-up visit, date of treatment admission, date of each follow-up visit, type of nutritional treatment received, description of treatment outcome at exit).

### Data cleaning

We respected each study's own final classification of children when they were discharged from the nutrition programme (study exit). We excluded children with oedema because their need to resolve fluid retention as part of their recovery adds a layer of complexity when handling MUAC- and weight- related indices. We also excluded follow-up visits of children which did not record any anthropometric measurements. At each visit, implausible height/length values (>120 cm or <60 cm), MUAC values (>200 mm or <70 mm), and weight values (<3.5 kg or >40 kg) were assigned a missing value. Z-scores were calculated for WHZ, WAZ, MUAC-for-age z-score (MUACZ) and height-for-age z-score (HAZ) based on the WHO (2006) growth standards [25], performed in R using the "zscorer" package [26]. Z-score outliers for WHZ, WAZ, MUACZ and HAZ were defined by a boxplot method [27] and assigned a missing value. Time points showing a negative difference of length/height between two visits were assigned a missing value. We defined implausible anthropometric differences between two weekly visits as weight>±1.5 kg, MUAC>±15 mm and height >1.5 cm.

 We included all children who were classified by their studies as recovered or as non-responders. Upon investigation of children with an 'unknown' outcome, we found their median length of follow- up to be 16 weeks and decided to re-classify them rather than

exclude them. As the criteria for anthropometric recovery in all but one of the studies were based on either reaching a MUAC ≥125mm or a WHZ ≥-2, children with either of those anthropometric parameters and without oedema upon their exit visit were re-classified as recovered, and those who had not reached either of those anthropometric criteria after at least 12 weeks of follow-up were re-classified as non-responders. The same procedure was used to re-classify defaulters. We excluded children classified as unknown or defaulters who had a length of stay less than 12 weeks. We excluded children who died or were transferred out of their treatment programme for medical reasons.

## Data analyses

We summarised the descriptive statistics of age, sex and anthropometric indices of children on study admission and exit. Anthropometric indices included weight, MUAC, height, WHZ, HAZ, WAZ, MUACZ, concurrent wasting and stunting (WaSt) and degree of wasting. For the latter, we defined severe wasting without oedema as WHZ <-3 or MUAC <115mm. We defined moderate wasting as WHZ ≥ -3 and <-2, or MUAC ≥ 115mm and < 125mm. We defined those children with WaSt as having both WHZ<-2 and HAZ<-2. We calculated the change in several anthropometric indicators between study admission and exit: weight gain (absolute and g/kg/day), and change in MUAC, weight, height, WHZ, WAZ, HAZ and MUACZ. We also summarised the length of stay in the nutritional programme in weeks, and caretaker reported morbidity at any point in the study (diarrhoea, cough, fever, vomiting and/ or rash, malaria). We present binary and categorical variables as N (%) and continuous variables as median (interquartile range; IQR).

First, we compared the above variables between the recovered children and the non-responders, testing differences between the two groups using a chi-squared test for binary and categorical variables and a Kruskal-Wallis test to compare continuous variables. We found that for several anthropometric outcomes (e.g., absolute MUAC gain, absolute WHZ gain) the median values between the two groups were similar although the IQRs differed. Intriguingly, the IQR of the non-responders showed that children in the first quartile were not growing at all (e.g., absolute gain near zero), but that three-quarters of the children were indeed growing. We therefore split the non-responder group into two further categories: low growth non-responders ('low growth NR') and high growth non-responders ('high growth NR'). We defined these two groups using thresholds derived from absolute MUAC, absolute weight, WAZ, WHZ and daily weight gain rate. For each of these anthropometric outcomes we defined low growth NR as children whose anthropometric gain between admission and exit was < 25th percentile (first quartile) for non-responders' pooled data. We defined high growth NR as children who gained ≥ 25th percentile. For example, the 25th percentile of MUAC gain from admission to exit for non-responders was 2mm, so the low growth NR group for this outcome included children with a MUAC gain <2mm and the high growth NR group included children with a MUAC gain ≥2mm.

We plotted the growth of the recovered, high growth and low growth NR groups over time, using mixed effects generalised additive models that accounted for clustering at the individual level. We plotted the growth trajectories of MUAC, weight, WHZ, WAZ, HAZ and height. We fitted models with a smoothing term using penalised regression splines and 95% confidence intervals. Covariates included the three exit categories (recovered, high growth NR, low growth NR) and interaction terms between the exit category and time. For growth trajectories of MUAC and weight we adjusted for the child's age at admission. We ran separate models to show the results for each definition of the high growth NR and low growth NR groups (i.e., using the 25th percentile threshold of gain in MUAC, weight WAZ, WHZ and daily weight

gain, as described above). We used the "mgcv" R package [28] to perform the models and the "itsadug" R package [29] to represent the modelled curves.

Given similarity of growth curves regardless of which anthropometric threshold was used to define the high growth NR and low growth NR groups, we continued our explorations just using the MUAC gain threshold. We chose MUAC gain as a practical indicator for further exploration as this anthropometric measurement is very commonly used in CMAM programmes, and sometimes used as the stand-alone admission and discharge criterion.

We compared age, sex and anthropometric characteristics of the recovered, high growth NR and low growth NR groups. For each variable we tested the difference between the high growth NR vs. recovered group and the low growth NR vs. recovered groups using the Wald test from univariate multinomial logistic regression. We explored the growth trajectories of the three categories of children according to whether the children had been admitted with severe wasting or moderate wasting.

Finally, to explore whether it was possible to predict which children were likely to be high growth NR or low growth NR versus recovered, we created a multivariate multinomial logistic regression model. This model compared the odds of being in the high growth NR or low growth NR groups at study exit, compared to being recovered, adjusting for all included variables. After assessing variables for multicollinearity by using the variance inflation factor (VIF; ensuring VIF <1.5), we included the following variables: sex, age category, MUAC category at admission, WHZ category at admission, WAZ category at admission, reported morbidity at any point in the study follow-up, no MUAC gain in first month, and no weight gain in first month. We checked the model accuracy and the area under the curve using the "tidymodels" R package [30].

All analyses were performed with R software and packages, version 4.2.3 [31]

## Ethical considerations

All datasets contributing to the pooled analyses came from research studies where ethical approval had been originally sought and granted, where principal investigators had agreed with a data sharing agreement for datasets to be included in the analyses for this paper, and where principal investigators confirmed that all planned analyses were within scope of the original ethical clearance. Full details of ethical clearance for all 14 studies are provided in S1 Table. All data was anonymised by individual principal investigators prior to sharing the data for compilation in the pooled dataset. The statistician (CC) was the only one with access to the pooled dataset and had no access to information that could identify individual participants throughout data analysis.

## Results

In **Fig 1** we show the flowchart for children included in this analysis. Starting with a total of 22,685 children from the 14 contributing studies, we excluded 4,631 children with oedema at admission and excluded a further 2,084 children who had either died, been transferred out of their studies, or had been in the defaulter or unknown category with a length of stay <12 weeks. This provided us with 15,970 children for the analyses. Using the original study definitions, after excluding children with oedema, 9.6% of children were categorised as non-responders.

We provide a detailed breakdown of individual study characteristics and outcomes in S1 and S2 Tables. In S3 Table we show overall, and by recovered versus non-responder categories, the age, sex and anthropometric characteristics of children on study admission, anthropometric characteristics on exit from the study, study follow-up characteristics, and anthropometric

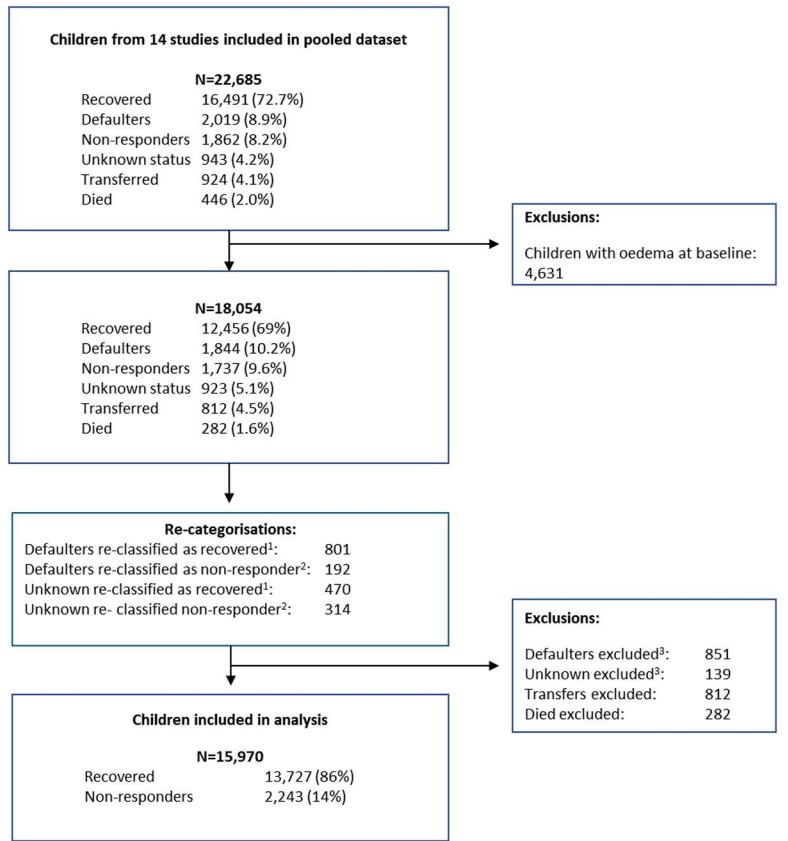

**Fig 1. Flow chart.** [1]For those children re-categorised, recovered refers to children who had at least one anthropometric criterion above the threshold of the WHO definition of acute malnutrition (MUAC ≥125mm or WHZ ≥−2) at the point of discharge from the nutritional programme, alongside the absence of any other criterion of severe wasting (i.e., MUAC≥115mm and WHZ≥-3 and no oedema), with a minimum length of stay of 12 weeks. [2]For those children re-categorised, non-responder refers to children who had both MUAC and WHZ below the threshold of the WHO definition of acute malnutrition (MUAC <125mm and WHZ <−2) at the point of discharge from the nutritional programme, with a minimum length of stay of 12 weeks. [3]Defaulters and unknown status with length of stay <12 weeks.

changes between study admission and exit. Compared to children who recovered, non-responders were slightly younger, and had a higher proportion of those with severe wasting and WaSt on admission. Non-responders had a worse WHZ, WAZ, MUAC, MUACZ on enrolment than those who recovered. The same proportion of non-responders and recovered children reported at least one morbidity over the follow-up.

**Characteristics of high growth and low growth non-responders.** In **Table 1** we show descriptive statistics and univariate multinomial logistic regression results for the three groups of children. Compared with children who recovered, the high growth NR group were younger and were admitted into the programme in a worse nutritional status for all anthropometric parameters except HAZ. Over three-quarters of high growth NR were severely wasted on admission. High growth NR had a higher median gain in MUAC, weight, MUACZ, WAZ, HAZ and a similar median WHZ gain, but with a length of stay of about 10 weeks longer. They had a lower weight gain rate but a higher absolute height and HAZ gain. Both groups showed a similar level of morbidity. In the low growth NR group, half of the children were severely wasted and half were moderately wasted on admission. Compared to children who recovered, children in the low growth NR group were slightly older. They had a marginally

**Table 1. Individual and nutritional programme characteristics between recovered, partial responders and treatment failures at enrolment, exit and over follow-up. Univariate multinomial regression analysis using 25th percentile of MUAC gain to define partial responders (MUAC ≥25th percentile; ≥2mm) and treatment failures (MUAC <25th percentile; <2mm).**

| Variable | N | Recovered N = 13,727 | High growth NR N = 513[1] | Low growth NR N = 718 | High growth non-responder OR | 95% CI | p-value* | Low growth non-responder OR | 95% CI | p-value* |
|---|---|---|---|---|---|---|---|---|---|---|
| **Sex** | 15,958 | | | | — | — | 0.012 | — | — | 0.012 |
| female | | 7,756 (57%) | 906 (60%) | 387 (54%) | — | — | | — | — | |
| Male | | 5,971 (43%) | 607 (40%) | 331 (46%) | 0.87 | 0.78, 0.97 | | 1.11 | 0.96, 1.29 | |
| **Age at baseline, months** | 15,958 | 17 (11, 28) | 14 (9, 26) | 18 (10, 36) | 0.99 | 0.99, 1.00 | <0.001 | 1.01 | 1.01, 1.02 | <0.001 |
| **Age at baseline, months** | 15,958 | | | | | | <0.001 | | | <0.001 |
| 6-11 | | 3,956 (29%) | 601 (40%) | 213 (30%) | — | — | | — | — | |
| 12-23 | | 5,111 (37%) | 456 (30%) | 219 (31%) | 0.59 | 0.52, 0.67 | | 0.80 | 0.66, 0.97 | |
| 24-59 | | 4,660 (34%) | 456 (30%) | 286 (40%) | 0.64 | 0.57, 0.73 | | 1.14 | 0.95, 1.37 | |
| **Acute malnutrition, at baseline (WHO definition)[1]** | 15,958 | | | | | | <0.001 | | | <0.001 |
| SAM | | 5,546 (40%) | 1,181 (78%) | 357 (50%) | — | — | | — | — | |
| MAM | | 8,181 (60%) | 332 (22%) | 361 (50%) | 0.19 | 0.17, 0.22 | | 0.69 | 0.59, 0.80 | |
| **Concurrently wasted and stunted at baseline, Yes[2]** | 14,647 | 6,182 (49%) | 711 (53%) | 363 (54%) | 1.18 | 1.05, 1.32 | <0.001 | 1.24 | 1.06, 1.45 | <0.001 |
| Unknown | | 1,089 | 173 | 49 | | | | | | |
| **Weight at baseline, kg** | 15,954 | 7.1 (6.3, 8.3) | 6.3 (5.5, 7.9) | 6.9 (6.0, 9.0) | 0.83 | 0.80, 0.86 | <0.001 | 1.02 | 0.98, 1.06 | <0.001 |
| Unknown | | 0 | 4 | 0 | | | | | | |
| **No weight gain in first month** | 15,958 | 3,665 (27%) | 639 (42%) | 384 (53%) | 2.01 | 1.80, 2.24 | <0.001 | 3.16 | 2.71, 3.67 | <0.001 |
| **Height/length at baseline, cm** | 14,674 | 73 (68, 80) | 70 (65, 79) | 73 (67, 85) | 0.99 | 0.99, 1.00 | <0.001 | 1.01 | 1.01, 1.02 | <0.001 |
| Unknown | | 1,067 | 169 | 48 | | | | | | |
| **MUAC at baseline, mm** | 15,941 | 120 (114, 122) | 111 (108, 116) | 120 (115, 120) | 0.89 | 0.88, 0.89 | <0.001 | 1.0 | 0.98, 1.01 | <0.001 |
| Unknown | | 17 | 0 | 0 | | | | | | |
| **MUAC, mm at baseline** | 15,941 | | | | — | — | <0.001 | — | — | <0.001 |
| <110 | | 1,102 (8%) | 420 (28%) | 58 (8%) | — | — | | — | — | |
| 110-114 | | 2,946 (21%) | 597 (39%) | 116 (16%) | 0.53 | 0.46, 0.61 | | 0.75 | 0.54, 1.03 | |
| 115-119 | | 2,422 (18%) | 239 (16%) | 169 (24%) | 0.26 | 0.22, 0.31 | | 1.33 | 0.98, 1.80 | |
| >=120 | | 7,240 (53%) | 257 (17%) | 375 (52%) | 0.09 | 0.08, 0.11 | | 0.98 | 0.74, 1.31 | |
| Unknown | | 17 | 0 | 0 | | | | | | |
| **No MUAC gain in first month** | 15,958 | 3,540 (26%) | 601 (40%) | 396 (55%) | 1.90 | 1.70, 2.12 | <0.001 | 3.54 | 3.04, 4.12 | <0.001 |
| **WHZ at baseline** | 14,668 | −2.4 (−3.4, −1.9) | −3.0 (−3.6, −2.4) | −2.9 (−3.4, −2.2) | 0.47 | 0.44, 0.50 | <0.001 | 0.62 | 0.56, 0.68 | <0.001 |
| Unknown | | 1,072 | 170 | 48 | | | | | | |
| **WHZ at baseline** | 14,668 | | | | — | — | <0.001 | — | — | <0.001 |
| >=−3 | | 9,609 (76%) | 665 (50%) | 395 (59%) | — | — | | — | — | |
| <−3 | | 3,046 (24%) | 678 (50%) | 275 (41%) | 3.22 | 2.87, 3.61 | | 2.20 | 1.87, 2.58 | |
| Unknown | | 1,072 | 170 | 48 | | | | | | |
| **MUAC Z score at baseline** | 15,941 | −2.8 (−3.4, −2.3) | −3.5 (−4.1, −2.9) | −3.0 (−3.5, −2.5) | 0.43 | 0.40, 0.46 | <0.001 | 0.86 | 0.79, 0.94 | <0.001 |
| **MUAC Z score at baseline** | 15,941 | | | | — | — | <0.001 | — | — | <0.001 |
| >=−3 | | 7,884 (58%) | 452 (30%) | 368 (51%) | — | — | | — | — | |
| <−3 | | 5,826 (42%) | 1,061 (70%) | 350 (49%) | 3.18 | 2.83, 3.56 | | 1.29 | 1.11, 1.50 | |
| Unknown | | 17 | 0 | 0 | | | | | | |
| **WAZ at baseline** | 15,954 | −3.2 (−3.9, −2.6) | −3.7 (−4.4, −2.9) | −3.5 (−4.1, −2.8) | 0.65 | 0.61, 0.68 | <0.001 | 0.81 | 0.75, 0.87 | <0.001 |

*(Continued)*

**Table 1.** (Continued)

| Variable | N | Recovered N = 13,727 | High growth NR N = [1]513 | Low growth NR N = [1]718 | High growth non-responder OR | 95% CI | p-value* | Low growth non-responder OR | 95% CI | p-value* |
|---|---|---|---|---|---|---|---|---|---|---|
| Unknown | | 0 | 4 | 0 | | | | | | |
| **WAZ at baseline** | 15,954 | | | | | | <0.001 | | | <0.001 |
| >=−3 | | 5,571 (41%) | 444 (29%) | 224 (31%) | — | — | | — | — | |
| <−3 | | 8,156 (59%) | 1,065 (71%) | 494 (69%) | 1.64 | 1.46, 1.84 | | 1.51 | 1.28, 1.77 | |
| Unknown | | 0 | 4 | 0 | | | | | | |
| **HAZ at baseline** | 14,639 | −2.7 (−3.6, −1.8) | −2.6 (−3.8, −1.5) | −2.7 (−3.7, −1.6) | 1.02 | 0.98,1.06 | 0.5 | 1.00 | 0.95, 1.05 | 0.5 |
| Unknown | | 1,095 | 175 | 49 | | | | | | |
| **HAZ at baseline** | 14,639 | | | | | | >0.9 | | | >0.9 |
| >=−3 | | 7,583 (60%) | 804 (60%) | 399 (60%) | — | — | | — | — | |
| <−3 | | 5,049 (40%) | 534 (40%) | 270 (40%) | 1.00 | 0.89, 1.12 | | 1.02 | 0.87, 1.19 | |
| Unknown | | 1,095 | 175 | 49 | | | | | | |
| **Morbidity, yes[3]** | 13,358 | 6,889 (59%) | 719 (61%) | 394 (77%) | 1.10 | 0.98, 1.25 | <0.001 | 2.26 | 1.84, 2.78 | <0.001 |
| Unknown | | 2,055 | 342 | 203 | | | | | | |
| **Hospitalisation** | 15,958 | | | | | | <0.001 | | | <0.001 |
| Hospitalised | | 408 (3.0%) | 138 (9.1%) | 56 (7.8%) | — | — | | — | — | |
| Not hospitalised | | 2,698 (20%) | 226 (15%) | 49 (6.8%) | 0.25 | 0.20, 0.31 | | 0.13 | 0.09, 0.20 | |
| Missing | | 10,621 (77%) | 1,149 (76%) | 613 (85%) | 0.32 | 0.26, 0.39 | | 0.42 | 0.31, 0.56 | |
| **Length of stay, week** | 15,958 | 6 (4 to 10) | 16 (13 to 18) | 14 (12 to 16) | 1.33 | 1.31, 1.35 | <0.001 | 1.29 | 1.27, 1.31 | <0.001 |
| **Weight gain, g/kg/day** | 15,797 | 2.4 (1.5 to 4.0) | 1.3 (0.9 to 1.9) | 0.6 (0.0 to 1.0) | 0.60 | 0.57, 0.62 | <0.001 | 0.37 | 0.34, 0.39 | <0.001 |
| Unknown | | 155 | 5 | 1 | | | | | | |
| **Weight gain, kg** | 15,945 | 0.8 (0.5 to 1.2) | 1.0 (0.7 to 1.3) | 0.4 (0.0 to 0.8) | 1.37 | 1.26, 1.48 | <0.001 | 0.21 | 0.18, 0.25 | <0.001 |
| Unknown | | 9 | 4 | 0 | | | | | | |
| **Length/height gain, cm** | 14,064 | 0.5 (0.0 to 1.1) | 1.0 (0.2 to 2.9) | 1.0 (0.0 to 2.5) | 1.47 | 1.42, 1.52 | <0.001 | 1.40 | 1.34, 1.46 | <0.001 |
| Unknown | | 1,523 | 293 | 78 | | | | | | |
| **MUAC gain, mm** | 15,922 | 8.0 (4.0 to 12.0) | 10.0 (6.0 to 12.0) | 0.0 (−2.0 to 1.0) | 1.04 | 1.03, 1.05 | <0.001 | 0.76 | 0.75, 0.77 | <0.001 |
| Unknown | | 36 | 0 | 0 | | | | | | |
| **WHZ gain** | 13,736 | 1.0 (0.5 to 1.6) | 1.0 (0.4 to 1.6) | 0.1 (−0.2 to 0.8) | 0.89 | 0.83, 0.96 | <0.001 | 0.25 | 0.22, 0.28 | <0.001 |
| Unknown | | 1,771 | 349 | 102 | | | | | | |
| **MUAC Z gain** | 15,742 | 0.8 (0.4 to 1.2) | 1.0 (0.7 to 1.3) | 0.0 (−0.2 to 0.1) | 1.59 | 1.47, 1.71 | <0.001 | 0.05 | 0.04, 0.06 | <0.001 |
| Unknown | | 201 | 10 | 5 | | | | | | |
| **WAZ gain** | 15,818 | 0.8 (0.5 to 1.2) | 1.1 (0.7 to 1.5) | 0.5 (0.0 to 0.7) | 1.63 | 1.51, 1.77 | <0.001 | 0.17 | 0.14, 0.20 | <0.001 |
| Unknown | | 100 | 32 | 8 | | | | | | |
| **HAZ gain** | 13,805 | 0.2 (0.0 to 0.4) | 0.4 (0.1 to 1.0) | 0.3 (0.0 to 0.9) | 3.39 | 3.08, 3.73 | <0.001 | 2.73 | 2.40, 3.09 | <0.001 |
| Unknown | | 1,723 | 335 | 95 | | | | | | |

Data are n (%) or median (IQR). CI, confidence interval; HAZ, height/length-for-age z score; IQR, interquartile range; MUAC, mid-upper-arm circumference; WAZ, weight-for-age z score; WHZ, weight for-height/length z score.

* p values show the Wald test result from univariate multinomial logistics regression. Odds ratio >1 indicate increased odds of being in either the high growth non-responder or low growth non-responder group compared to being recovered.

[1] WHO definition of severe acute malnutrition (SAM) is defined as MUAC<115 mm or WHZ <−3 or oedema. Children with oedema or with missing data on oedema are excluded from this analysis. Moderate acute malnutrition (MAM) is defined as MUAC ≥115 mm and < 124mm or WHZ ≥3 and <−2, with no oedema.

[2] Concurrently wasted and stunted (HAZ <−2 and WHZ <−2)

[3] Morbidity includes diarrhoea, cough, fever, vomiting and/or rash, malaria at inclusion or during follow-up.

worse nutritional status at admission by WHZ, MUACZ, and WAZ, but a similar median MUAC and HAZ. Although their growth in all indices involving weight and MUAC was minimal, they displayed a higher absolute height and HAZ gain, and reported greater morbidity.

## Growth curves

When comparing anthropometry (MUAC, weight, WHZ, and WAZ) over the duration of treatment, growth between those considered recovered and those considered non-responders showed a near parallel trajectory (**Fig 2**). The main difference between the two groups was that non-responders started with lower anthropometric values at admission (S3 Table).

**Fig 3** compares the growth trajectories of children's MUAC, weight, WHZ and WAZ over the treatment duration disaggregated into three groups: 1) recovered children, 2) high growth NR, and 3) low growth NR. After disaggregating out those with the least amount of growth (i.e., the low growth NR), trajectories of the high growth NR remained in parallel to those that recovered in the first half of treatment, with a slightly smaller gap between the two groups in the second half of treatment, signifying slightly faster catch-up growth among this group of high growth NR. In contrast, the low growth NR trajectory was distinct from both the recovered and high growth NR groups, showing limited growth throughout treatment. Notably, the low growth NR group started with moderate wasting by MUAC and WHZ on admission but ended with the lowest anthropometry among the three groups upon discharge.

Growth curves where high growth NR and low growth NR were defined by the 25th percentile of median absolute weight gain (S1 Fig A-D), median daily weight gain rate (S2 Fig A-D), median WAZ gain (S3 Fig A-D), and median WHZ gain (S4 Fig A-D) show similar patterns to those described above.

Trajectories of HAZ and absolute height gain over time, with the high growth NR and low growth NR groups defined by MUAC gain and daily weight rate gain thresholds (S5 Fig A-D), and by WHZ and WAZ gain thresholds (S6 Fig A-D) demonstrate all groups increased in absolute height and HAZ over time. When using the thresholds of MUAC gain, daily weight gain rate, and WAZ gain, there was no difference in trajectories of HAZ and absolute height between the three groups. When classifying groups using the WHZ threshold, the low growth NR group experienced a slightly faster rate of HAZ gain from week 10 onwards.

We show the growth trajectories of the three groups of children according to whether they were admitted with severe wasting (**Fig 4**) or moderate wasting (**Fig 5**). In each scenario the growth trajectories of the three groups were similar to as described above (Fig 3).

For children with severe wasting on admission, the high growth NR took two months for their mean MUAC to cross ≥115mm, and two weeks for their WHZ to cross ≥-3. It took them over 3 months of treatment to reach WAZ ≥-3. The low growth NR group remained severely wasted and very severely underweight (mean WAZ ≤ -3.5) at discharge. For children with moderate wasting on admission the high growth NR increased their MUAC, weight, WHZ and WAZ steadily over the whole treatment time but had a slightly slower growth rate than the recovered children in the first 3-4 weeks and almost reached standard recovery cutoffs of WHZ and MUAC by week 14. Low growth NR remained moderately wasted and severely underweight throughout their treatment.

**Predictors of being in the high growth or low growth NR groups.** Regarding potential predictors of being in the different groups (**Fig 6**), there were three co-variates that differentiated between the risk of being in the low growth NR group compared to being in the high growth NR group (i.e., factors resulting in different directions of effect sizes between the high growth NR and low growth NR groups, compared to children who recovered). Firstly, sex of child: compared to the recovered children, boys had reduced odds

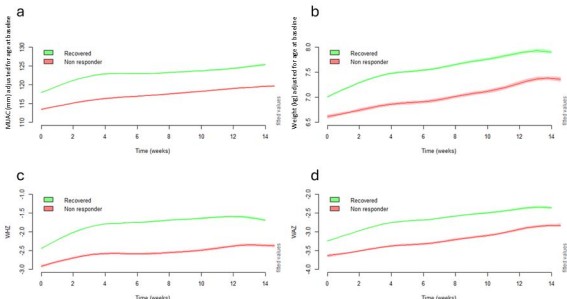

**Fig 2. Panel of modelled adjusted weekly means of a) MUAC, b) weight, c) WHZ and d) WAZ over time comparing children classified as recovered with non-responders.** MUAC, mid-upper-arm circumference; WAZ, weight-for-age z-score; WHZ, weight-for-height z-score.

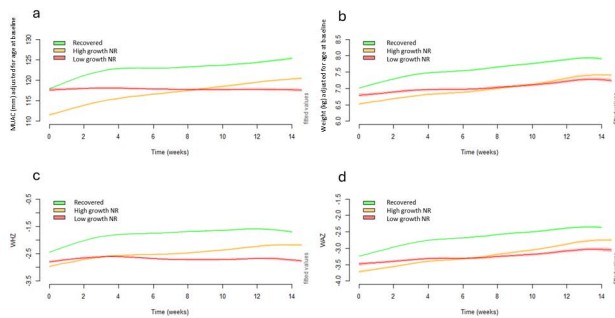

**Fig 3. Panel of modelled adjusted weekly means of a) MUAC, b) weight, c) WHZ and d) WAZ over time using 25th percentile of MUAC gain to define high growth non-responders (MUAC ≥25th percentile; ≥2mm) and low growth non-responders (MUAC <25th percentile; <2mm).** MUAC, mid-upper-arm circumference; WAZ, weight-for-age z-score; WHZ, weight-for-height z-score.

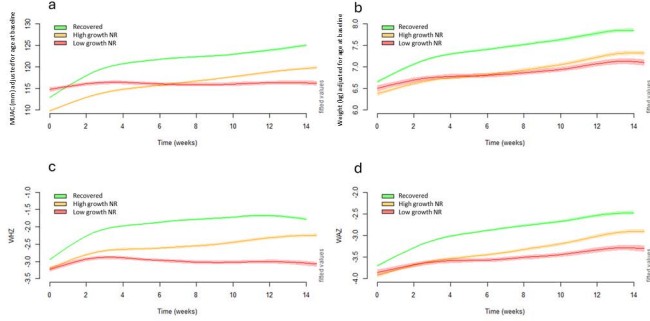

**Fig 4. Panel of modelled adjusted weekly means of a) MUAC, b) weight, c) WHZ and d) WAZ over time using 25th percentile of MUAC gain to define high growth non-responders (MUAC ≥25th percentile; ≥2mm) and low growth non-responders (MUAC <25th percentile; <2mm) in children with severe wasting[1]at baseline.** MUAC, mid-upper-arm circumference; WAZ, weight-for-age z-score; WHZ, weight-for-height z-score. [1]Severe wasting defined as WHZ <-3 or MUAC <115mm.

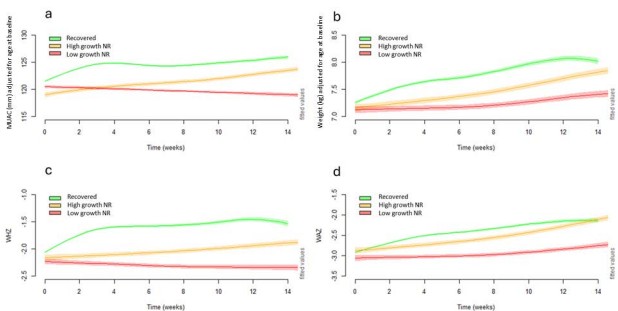

**Fig 5. Panel of modelled adjusted weekly means of a) MUAC, b) weight, c) WHZ and d) WAZ over time using 25th percentile of MUAC gain to define high growth non-responders (MUAC ≥25th percentile; ≥2mm) and low growth non-responders (MUAC <25th percentile; <2mm) in children with moderate wasting[1] at baseline.** MUAC, mid-upper-arm circumference; WAZ, weight-for-age z-score; WHZ, weight-for-height z-score. [1] Moderate wasting defined as WHZ ≥ -3 and <-2, or MUAC ≥ 115mm and < 125mm.

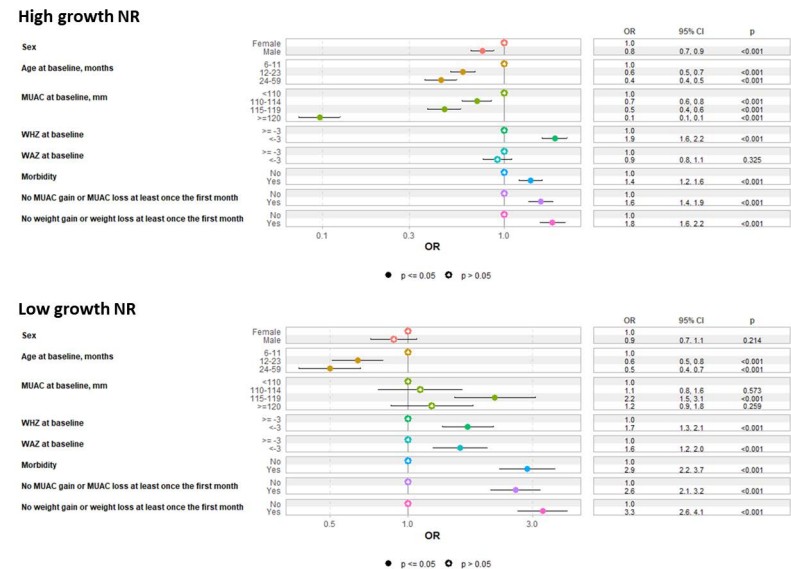

**Fig 6. Multivariate multinomial logistic regression using 25th percentile of MUAC gain to define high growth non-responders (MUAC ≥25th percentile; ≥2mm) and low growth non-responders (MUAC <25th percentile; <2mm).** Reference group are the recovered children. Odds Ratio >1 indicate increased odds of being in either the high growth non-responder or low growth non-responder group compared to being recovered. P values show the Wald test result after all variables in model adjusted for. MUAC, mid-upper-arm circumference; WHZ, weight for-height/length z-score; WAZ=weight-for-age z score. Morbidity includes diarrhoea, cough, fever, vomiting and/or rash, malaria at inclusion or during follow-up.

of being in the high growth NR group relative to girls. However, there was no difference by sex in the odds of being in the low growth NR group. Secondly, MUAC at admission: compared to the recovered group, higher MUAC categories at admission reduced the odds of being in the high growth NR group, relative to the lowest MUAC category (<110mm). In contrast, MUAC category at admission made no difference to the odds of being in the low growth NR group, apart from having a MUAC of 115-119 mm, which increased the odds relative to the MUAC <110mm category. Thirdly, WAZ at admission: compared to the

children who recovered, being severely underweight (WAZ <-3) at admission increased the odds of being in the low growth NR group, relative to children with WAZ ≥ -3. Yet there was no difference in the odds of being in the high growth NR group by WAZ category at admission.

All the other model co-variates did not specifically differentiate between the risk of being in the high growth NR or low growth NR groups, compared to recovered children. However, the following co-variates were all associated both with increased odds of being in the high growth NR group and increased odds of being in the low growth NR group compared to recovered children: WHZ <-3 vs. WHZ ≥ -3, any history of morbidities vs. none, no MUAC gain in the first month, and no weight gain in the first month. Compared to the youngest age bracket (6-11 months), older children had reduced odds of being in both the high growth NR and low growth NR groups compared to recovered children. Overall, the model accuracy was 87.7% and had an area under the curve of 74.1% (**Fig 7**).

## Discussion

### High growth non-responders are treatment successes rather than failures

Our data show that 'non-responders' are not a homogeneous group and illustrate the need to differentiate between those who experience minimal growth over the treatment duration (low growth NR group) versus those who are responding to treatment, as shown by the high growth NR group. Indeed, this latter group should be viewed as treatment successes, since their growth curves show they are growing just as well as the children who recover, following the same trajectory, but starting with a worse anthropometric status. At the group level, we did not see any evidence to suggest children categorised as high growth NR had reached the limits of their growth.

Given we divided the original non-responders into high growth NR and low growth NR using the 25th percentile of various anthropometric outcomes, approximately 75% of 'non responders' seem to be responding to treatment very well. By regarding them as 'non-responders' instead of children who are responding well to treatment, we potentially risk discharging them too early, as well as underestimating the overall effectiveness of therapeutic feeding programmes.

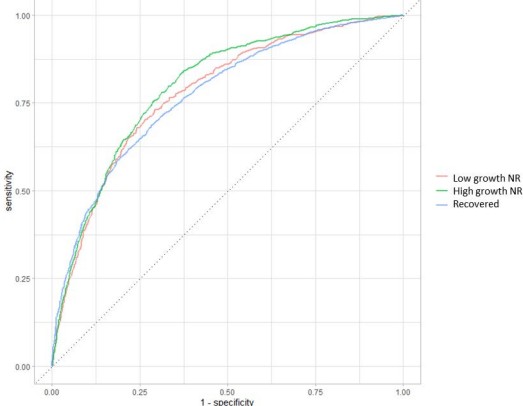

**Fig 7. Area under the ROC of the multivariate multinomial logistic regression model corresponding to Fig 6.** Model accuracy = 87.7%. Area under the ROC=74.1%.

## High growth non-responders may require longer to recover

Children who recovered started to slow their rate of growth of MUAC, WAZ and WHZ after 3-4 weeks of treatment and then gradually began to plateau around the common thresholds of recovery for MUAC (125mm) and WHZ (-2 z-scores), and at approximately -2.5 z-scores for WAZ. Similar growth patterns have been previously documented for MUAC [22,32], WAZ [20,32] and WHZ [20]. This supports the rationale that gradual plateauing after a period of growth may be an indicator of appropriate discharge, given that we do not see significant growth gains thereafter.

Using this logic, we propose that the high growth NR group may need to be kept longer under treatment to enable them to chance to reach a plateau in their growth, given there was no indication of them reaching a growth plateau in our data. Looking at the MUAC growth trajectory of the high growth NR group as an example (Fig 3A), children admitted with a very low MUAC (i.e., ~110mm) would likely take longer than four months to cross the MUAC threshold for recovery (>125mm). Given the chance to remain longer under treatment, it is possible many children in the high growth NR group would go on to reach the standard discharge thresholds for MUAC and/or WHZ, but this requires further research to verify.

## Low growth non-responders are the true 'non-responders'

It is the low growth NR group who are the true 'non-responders', as their growth curves demonstrated how no matter how long they stayed under treatment they did not improve their MUAC- or weight- related indices. This group had the highest proportion of children with reported co-morbidities. In multivariate analyses, being severely underweight (WAZ <-3) on admission was associated with increased odds of being in the low growth NR group compared to recovered children, yet this was not associated with the odds of being in the high growth NR group. It is therefore possible that many of these children have constrained growth because they are sick and/or have a disability. As the relationships between environmental enteric dysfunction, impaired gastrointestinal tract function, microbiota immaturity, and therapeutic responses remain unclear, we also cannot exclude the possibility that these factors may also be drivers of poor treatment response [33,34]. It is important to note, however, that these characteristics might not necessarily extend to the whole group. For example, some children in the low growth NR group include those admitted with relatively high anthropometry (just below the cut-off for being moderately wasted) and therefore might not be expected to grow much more in their weight or MUAC indices. Nonetheless, we cannot conclude that non-response in weight and MUAC indices equates to a total lack of benefit of the therapeutic treatment, as our current analysis does not include data on other outcomes such as micronutrient status, immune function or cognitive development.

On the latter point, it is intriguing that those in the low growth NR group as defined by the lowest WHZ gain quartile were those who gained height the best but seemingly at the expense of weight gain. This is shown in S4 Fig C where we see a noticeable decline in WHZ after week 4, corresponding to the improved HAZ in that group in S6 Fig A. Previous research demonstrates the complex relationships between nutrition and linear growth [35], where the opposite pattern has been seen with linear growth following gains in WHZ [36,37]. Further investigation would be needed to explore whether the height gain is a consistent finding, and if so, whether it could be because the children were not too severely wasted on admission, or whether measurement error in height could play a role in explaining the finding.

## Difficulty in predicting on admission how children will respond to treatment

The growth curves show that it is not easy to predict which children will go on to recover, be in the high growth NR group or in the low growth NR group, given growth trajectories cross between the high growth and low growth NR groups between weeks 4-8 depending on the anthropometric outcome. Severe underweight on admission was associated with increased odds of being in both the low growth NR and high growth NR groups compared to recovered children in univariate analyses (Table 1), whilst in multivariate analyses (Fig 6) severe underweight was only associated with increased odds of being in the low growth NR group. Therefore, whilst this indicator alone is insufficient to fully predict outcomes, it does mean health workers should be aware of the increased vulnerability of children who are severely underweight at admission. The heightened risk of mortality for children who are severely underweight has previously been described [38,39]. Various co-variates increased or decreased the odds of being in both the high growth and low growth NR groups, but did not differentiate between the two NR groups. For example, older children (age 12-59 months) had reduced odds of being in both NR groups compared to younger children (age 6-23 months). This is consistent with other research showing that younger children are more at risk of being non-responders [40], and may in part be because in MUAC-based programming younger children tend to have a smaller MUAC on admission than older children and take longer to reach the discharge criteria. Our model was constrained by the available variables in the datasets; further research may discover other characteristics that would enhance predictive power.

## Programmatic implications

Our findings underscore the importance of weekly monitoring of MUAC and/ or weight curves to assess growth trajectories. If there is consistent growth, yet the children have not yet reached the discharge criteria, one option would be to keep children on the programme until they either reach discharge criteria, or show signs of plateauing, even if this requires extending the length of stay beyond usual maximum cut-offs of 12-16 weeks. This seems particularly important for those children admitted with severe wasting by MUAC, as it takes them two months just to cross the cut-off from severe to moderate wasting (i.e., for their MUAC to recover to ≥115mm). Further implementation research would need to assess how best healthcare workers could determine when an appropriate growth plateau has been reached, considering in many health facilities growth charts are not plotted and/or not analysed. Additional training may be required to ensure staff can track those children showing signs of consistent growth but who fall short of reaching discharge criteria, and that these children are kept under treatment until no further growth is seen. Further research will be needed to provide guidance on the most user-friendly operational definitions of 'consistent growth' and 'no further growth'. Further operational research would also need to confirm whether growth monitoring at the individual level is realistic or whether a simpler approach should be tested where, for example, all children with MUAC <110mm or WAZ<-3 on admission should have a fixed, extended, length of time under treatment.

It may not be necessary for children whose length of stay in the programme is extended to continue receiving the same dosage of therapeutic food throughout; some of these children could potentially have their doses of therapeutic food safely tapered. Similarly, not all children may need weekly follow-up and could perhaps switch to fortnightly monitoring. Previous research suggests that stratifying children by risk level could provide a promising avenue for further exploration, for example, where children with a low MUAC (e.g., <120mm) and/or a low WAZ (e.g., <-3) continue to receive intensive treatment and follow up, whereas children

with higher MUAC and WAZ receive less intensive treatment [20]. The OptiMA trial in DRC has shown that progressively reducing the RUTF dose as children with severe wasting recover is non-inferior to standard RUTF dosage [22]. The new WHO guidelines also indicate that RUTF dosage can be reduced once the child is no longer severely wasted [6].

Keeping children under treatment for longer may have cost implications. However, given that in our pooled dataset 9.6% of children were originally categorised as non-responders (Fig 1) and 75% of these were furthered categorised as high growth NR, we might only expect ~7% of all admissions to require longer treatment to recover. Further research would be needed to ascertain if keeping a child for some extra weeks to ensure a more complete recovery (whether reaching discharge thresholds or plateauing growth) reduces the risk of relapse, which might save programme costs by reducing future re-admissions. Furthermore, the increased costs for this relatively small number of children who are treated longer are likely to be ultimately cost-effective by reducing these children's vulnerability to significant morbidity and mortality. This is plausible given that compared to children who have not been wasted, children who are discharged from CMAM programmes already have 3-5 times the risk of relapsing to being wasted or dying within 6-months post-discharge, and, furthermore, those with the lowest anthropometric status on discharge have the highest likelihood of relapsing [41].

The low growth NR group represents an even smaller proportion of children receiving treatment, in this dataset equating to around 2.4%. Current CMAM programme advice indicates that if a child does not gain weight for 2 or 3 consecutive weeks (at any point during the treatment), it is recommended to check the social context and the effective use of RUTF at home. Then, if the problem has not been resolved, children should be referred to hospital to explore other medical co-morbidities such as tuberculosis, HIV, sickle cell anaemia, congenital cardiopathy or neurological disease [6]. Our data aligns with this recommendation, but in addition suggests that children could be referred for further investigations early in the treatment programme, possibly as early as when no improvement in weight or MUAC is seen between weeks 3 and 4. The recent WHO guidelines allow for community health workers to manage wasting treatment programmes under adequate training and regular supervision (recommendation B17) [6]. One aspect of 'adequacy' in terms of that supervision could include a detailed case review of the first 4-5 weeks of treatment and adherence to referral protocols as appropriate.

The realities are that often medical referrals do not happen due to lack of transportation to hospitals, inappropriate medical facilities or diagnostics, or barriers due to cost of treatment. Screening for disabilities remains an area where feasible tools are still urgently required [42]. However, progress is being made in various diagnoses, e.g., WHO have recently recommended an interim conditional recommendation on using a 'treatment decision algorithm' to help with TB treatment decision making [43] and there is progress on use of rapid tests for sickle cell anaemia [44]. In the case of other non-medical determinants of low growth NR (e.g., young child feeding practices at household level), further investigations may be hindered by constrained staffing or capacities.

As discussed above, it is possible not all the children in the low growth NR group have clear underlying determinants, and some may be those admitted on the border of not being wasted. Practitioners should therefore take account of anthropometric status on admission, history of co-morbidities, accounts of appetite and alertness to discern whether to refer for more detailed medical investigations or not. Of note, however, is that in our dataset the low growth NR group were severely underweight throughout treatment, whether admitted with severe or moderate wasting. This would suggest this group remains at heightened risk of mortality if untreated [39]. For those children found in the low growth NR category who do not turn out to have underlying co-morbidities, we do not have sufficient evidence from our data to advise

them being retained in the programme, receiving lower dosages of therapeutic food, or being discharged early; all this requires further investigation.

In our analyses we used the 25th percentile of growth across a variety of anthropometric outcomes to define the high growth NR versus low growth NR groups. This was an analytical approach that enabled exploration of the heterogeneity of 'non responders' but does not necessarily require practitioners to re-create these categories within their therapeutic feeding programmes. In whichever way we defined the groups of low growth NR and high growth NR groups (i.e., 25th percentile of MUAC, weight, weight gain rate, WAZ, WHZ gain), the growth trajectories were similar. This means that simpler measures such as weight and/or MUAC can be monitored and there is no need for more complex indicators (e.g., WHZ, rate of weight gain) to be calculated if those were to be less feasible. What is relevant for practical implications is the importance of early referral for children who are not growing in MUAC or weight, especially after the first three weeks of treatment, and the continued monitoring of anthropometry so that children who are growing, but who have not met the discharge criteria yet, are kept in the programme until they do reach the thresholds or show signs of plateauing growth. For other researchers wanting to explore these categories of children in their own data, however, using the 25th percentile of the non-responders' growth was a helpful way to define two groups of high growth NR and low growth NR. Indeed, recent research into characteristics of non-responders in Niger found very similar patterns, whereby approximately 80% of non-responders had a similar rate of MUAC gain to those who recovered, and approximately 20% of non-responders showed no sign of MUAC improvement throughout treatment [40].

We recognise that the ability to detect growth or lack of growth, whichever anthropometric indicator is tracked, is dependent on the quality of measurements. Hence continued supervision of measurement technique and accurate data capture is critical. Finally, we cannot discount that some children in the high growth NR or low growth NR groups may have missed the opportunity to recover because programme quality was not good enough, as opposed to (or in addition to) co-morbidities or social factors.

## Limitations

The 14 datasets making up our pooled analyses were heterogenous, having different admission and discharge criteria, different time intervals between consecutive visits, different treatment regimes, and various maximum durations of treatment. We summarise the different potential study-level and individual-level effect modifiers of response to treatment by study in tables S1 and S2. Whilst our dataset is multi-country, it is not necessarily representative of all contexts where wasting is treated. The included datasets did not include contextual factors such as the nutrition status of the general population, or the type of nutritional shocks being experienced (e.g., endemic malnutrition versus sudden food insecurity shocks). These contextual factors could influence what proportion of children respond to treatment and at what point growth trajectories flatten – considerations that would be interesting to explore in further research. There were also potentially influential individual-level variables that the datasets did not all have, such as birthweight and infant feeding practices. Our analyses should therefore be considered exploratory, with the aim of generating hypotheses that we hope can be further tested operationally.

## Conclusions

Our exploratory analyses have shown that children categorised as 'non-responders' are not a homogenous group. Approximately 75% of non-responders are better viewed as 'delayed responders' who respond to treatment similarly to those who recover. For these delayed

responders, our data show a strong rationale for maintaining treatment in order to ensure growth potential has been met, with the aim of avoiding the additional vulnerabilities that can accompany early discharge. Only the smaller proportion of low growth NR, i.e., those failing to grow in MUAC and weight at all, should be viewed as true non-responders or treatment failures. The current practice whereby all non-responders are regarded as treatment failures risks underestimating the effectiveness of the CMAM model. Other practical implications of this research align well with current guidance, whereby all children should have their growth monitored regularly (weekly), at least for MUAC and weight. Our data suggest if after 3 weeks of treatment there are no improvements in growth, it is likely that there will be no further improvement, therefore appropriate referrals for further checks on chronic morbidities and social determinants should be made. Those running healthcare services will need to invest in system-wide resources for adequate screening, referral, diagnosis, and treatment of the multiple determinants of growth failure, critical components that are commonly absent. However, the precise implementation guidance of how to determine when a growth plateau has been reached, what the ideal intensity of additional treatment and follow-up is, and whether a final maximum length of stay can be defined, are all questions that remain to be answered with further operational research.

## Supporting information

**S1 Table. Descriptive information on potential study-level effect modifiers, by study.** (XLSX)

**S2 Table. Descriptive information on potential individual-level effect, by study** (XLSX)

**S3 Table. Individual and nutritional programme characteristics between recovered and non-responders, at enrolment, exit and over follow-up.** (DOCX)

**S1 Fig. Panel of modelled adjusted weekly means of a) MUAC, b) weight, c) WHZ and d) WAZ over time using 25th percentile of absolute median weight gain to define high growth non-responders (absolute median weight gain ≥25th percentile; ≥0.4 kg) and low growth non-responders (absolute median weight gain <25th percentile; <0.4 kg).** MUAC, mid-upper-arm circumference; WAZ, weight-for-age z-score; WHZ, weight-for-height z-score. (TIF)

**S2 Fig. Panel of modelled adjusted weekly means of a) MUAC, b) weight, c) WHZ and d) WAZ over time using 25th percentile of absolute median daily weight gain to define high growth non-responders (absolute median daily weight gain ≥25th percentile; ≥0.6 g/kg/day) and low growth non-responders (absolute median daily weight gain <25th percentile; <0.6 g/kg/day).** MUAC, mid-upper-arm circumference; WAZ, weight-for-age z-score; WHZ, weight-for-height z-score. (TIF)

**S3 Fig. Panel of modelled adjusted weekly means of a) MUAC, b) weight, c) WHZ and d) WAZ over time using 25th percentile of absolute median WAZ gain to define high growth non-responders (absolute median WAZ gain ≥25th percentile; ≥0.5 z-score) and low growth non-responders (absolute median WAZ gain <25th percentile; <0.5 z-score).** MUAC, mid-upper-arm circumference; WAZ, weight-for-age z-score; WHZ, weight-for-height z-score. (TIF)

**S4 Fig. Panel of modelled adjusted weekly means of a) MUAC, b) weight, c) WHZ and d) WAZ over 16 weeks using 25th percentile absolute median WHZ gain to define high growth non-responders (absolute median WHZ gain ≥25th percentile; ≥0.0 z-score) and low growth non-responders (absolute median WHZ gain <25th percentile; <0.0 z-score).** MUAC, mid-upper-arm circumference; WAZ, weight-for-age z-score; WHZ, weight-for-height z-score.
(TIF)

**S5 Fig. Modelled adjusted weekly means of HAZ and absolute height over time. High growth non-responders and low growth non-responders defined using 25th percentile of MUAC gain (panel a, b) and absolute daily weight gain (panel c, d).** MUAC, mid-upper-arm circumference; WAZ, weight-for-age z-score; WHZ, weight-for-height z-score.
(TIF)

**S6 Fig. Modelled adjusted weekly means of HAZ and absolute height over time. High growth non-responders and low growth non-responders defined using 25th percentile of absolute median WHZ gain (panel a, b) and absolute median WAZ gain (panel c, d).** MUAC, mid-upper-arm circumference; WAZ, weight-for-age z-score; WHZ, weight-for-height z-score.
(TIF)

## Acknowledgments

We thank the families and children who consented to have their data included in the original research studies and the staff who worked hard to collect this data and provide the treatment for wasting. We also thank the primary investigators and institutions who generously shared their data to contribute to this pooled analysis: Mark Manary, Washington University in St Louis; Crystal Karakochuk and Stanley Zlotkin, Hospital for Sick Kids in Toronto; Sunita Taneja, SAS India; Kate Sadler at Valid International, Susan Shepherd at the Alliance for International Medical Action (ALIMA), Kevin Phelan at ALIMA, and Renaud Becquet, University of Bordeaux, National Institute for Health and Medical Research (Inserm), Research Institute for Sustainable Development (IRD), and Bordeaux Population Health Center, Bordeaux, France. Thank you to the academic editors, Dr Dickson Abanimi Amugsi and Dr Gerard Bryan Gonzales; the named reviewers, Prof. Jay Berkley and Dr Laura C Altobelli; and the anonymous reviewer for their comments, which strengthened this manuscript.

## Author contributions

**Conceptualization:** Cécile Cazes, Heather Stobaugh, Tanya Khara, Philip T. James.

**Data curation:** Heather Stobaugh, Tanya Khara.

**Formal analysis:** Cécile Cazes.

**Funding acquisition:** Tanya Khara.

**Investigation:** Cécile Cazes, Indi Trehan, Philip T. James.

**Methodology:** Cécile Cazes, Heather Stobaugh, Indi Trehan, Philip T. James.

**Supervision:** Tanya Khara, Philip T. James.

**Writing – original draft:** Cécile Cazes, Philip T. James.

**Writing – review & editing:** Cécile Cazes, Heather Stobaugh, Paluku Bahwere, Paul Binns, Robert E Black, Erin Boyd, André Briend, Sheila Isanaka, Suvi T Kangas, Tanya Khara, Natasha Lelijveld, Martha Mwangome, Mark Myatt, Gloria Odei Obeng-Amoako, Indi Trehan, Philip T. James.

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
