## [Decision Letter · Decision Letter 0]

30 Jul 2024

PGPH-D-24-01377

Re-thinking “non-response” to wasting treatment: exploratory analysis from 14 studies

Dear Dr. James,

Thank you for submitting your manuscript to PLOS Global Public Health. After careful consideration, we feel that it has merit but does not fully meet PLOS Global Public Health’s publication criteria as it currently stands. Therefore, we invite you to submit a revised version of the manuscript that addresses the points raised during the review process.

Additionally, I encourage the authors to strongly consider the comments of Reviewer 1 about the dichotomisation of the growth trajectories.

We look forward to receiving your revised manuscript.

Kind regards,

Gerard Bryan Gonzales

Academic Editor

Journal Requirements:

Additional Editor Comments (if provided):

Reviewers' comments:

Reviewer's Responses to Questions

**Comments to the Author**

1. Does this manuscript meet PLOS Global Public Health’s publication criteria ? Is the manuscript technically sound, and do the data support the conclusions? The manuscript must describe methodologically and ethically rigorous research with conclusions that are appropriately drawn based on the data presented.

Reviewer #1: Yes

Reviewer #2: Yes

2. Has the statistical analysis been performed appropriately and rigorously?

Reviewer #1: No

Reviewer #2: Yes

3. Have the authors made all data underlying the findings in their manuscript fully available (please refer to the Data Availability Statement at the start of the manuscript PDF file)?

Reviewer #1: Yes

Reviewer #2: Yes

4. Is the manuscript presented in an intelligible fashion and written in standard English?

Reviewer #1: Yes

Reviewer #2: Yes

5. Review Comments to the Author

Reviewer #1: This thought-provoking paper deals with the important question of how we define non-response in children being treated for SAM/severe malnutrition/wasting (nomenclature in flux). The authors have assembled a dataset of 14 studies and reanalysed their data, representing 22,685 children but excluding 4,631 with oedematous malnutrition. The manuscript is well written and clear.

I have one reservation. The whole analysis rests on a comparison of the growth trajectories dichotomised into groups above and below the 25% centile of anthropometric gain during follow up. The authors conclude that the upper 75% actually grow fairly well, but the bottom 25% don’t. Without a sensitivity analysis, showing that other dichotomies do not discriminate so well between good and poor growth trajectories, the authors have merely generated a very sophisticated way of showing that children growing well grow well, and that children growing badly grow badly. I would like to see evidence that dichotomising around 40% or 15% is less effective at predicting poor recovery. This would also make the ROC analysis less arbitrary. The methodological explanation given in lines 125-127 is not really very helpful, but the authors do acknowledge the difficulty in lines 333-335. A sensitivity analysis would go a long way to resolving this.

Minor points

Figure 1 – should be “adjusted for”, not “adjusted on”.

Lines 335-6 – “self-reported morbidities” is surely inappropriate for children.

Lines 337-387 – isn’t it self-evident that very malnourished children are more likely to take time to respond and have a higher risk of non-response?

Reviewer #2: This is an important topic, a novel approach to thinking about it and is valuable to the SAM treatment community. The analyses appear appropriate. I have only relatively minor comments that may improve the manuscript.

In the Abstract, a couple of things were unclear or ambiguous:

• It would be helpful to indicate whether the 25th percentile gain is with reference to the study populations(s) or WHO standards.

• ‘For all outcomes, the high growth non-responders tracked along a near-parallel growth trajectory to children who recovered but started with lower anthropometry’ – does this mean that the non-responders or those who recovered had lower initial anthropometry?

At lines 128-130, there is the same issue of what the percentile refers to.

At line 134, ‘mixed effects’ could mean clustered by individual and/or site – please clarify. Likewise, at line 155, were shared characteristics within sites accounted for?

Might the context of the study be important? For example, endemic malnutrition vs. sudden food insecurity shock may mean different underlying deficiencies and responses of weight, height and MUAC to treatment. Could this be tested, e.g. in results presented in Table 1? At line 302 I was also wondering if this co-variate might help differentiate between the risk of being in the high growth NR or low growth NR groups. Presumably, the community norm might also influence when and at what level trajectories flatten.

At line 279, did age influence the time to cross MUAC 115mm? At age 6 months, infants may struggle more to reach a threshold than a 4 year old since population MUAC is not flat from 6-59 months.

At line 410, non-anthropometric individual risks like low birth weight, breast feeding etc could be mentioned, as they are mentioned as limitations of the datasets at line 432 (perhaps this should be under line 481).

6. PLOS authors have the option to publish the peer review history of their article (what does this mean? ). If published, this will include your full peer review and any attached files.

**Do you want your identity to be public for this peer review?** For information about this choice, including consent withdrawal, please see our Privacy Policy .

Reviewer #1: No

Reviewer #2: **Yes: ** James A Berkley

---

## [Decision Letter · Decision Letter 1]

15 Oct 2024

PGPH-D-24-01377R1

Re-thinking “non-response” to wasting treatment: exploratory analysis from 14 studies

Dear Dr. James,

Thank you for submitting your manuscript to PLOS Global Public Health. After careful consideration, we feel that it has merit but does not fully meet PLOS Global Public Health’s publication criteria as it currently stands. Therefore, we invite you to submit a revised version of the manuscript that addresses the points raised during the review process.

We look forward to receiving your revised manuscript.

Kind regards,

Dickson Abanimi Amugsi, PhD

Academic Editor

Journal Requirements:

Additional Editor Comments (if provided):

Reviewers' comments:

Reviewer's Responses to Questions

**Comments to the Author**

1. If the authors have adequately addressed your comments raised in a previous round of review and you feel that this manuscript is now acceptable for publication, you may indicate that here to bypass the “Comments to the Author” section, enter your conflict of interest statement in the “Confidential to Editor” section, and submit your "Accept" recommendation.

Reviewer #1: (No Response)

Reviewer #2: All comments have been addressed

Reviewer #3: (No Response)

2. Does this manuscript meet PLOS Global Public Health’s publication criteria ? Is the manuscript technically sound, and do the data support the conclusions? The manuscript must describe methodologically and ethically rigorous research with conclusions that are appropriately drawn based on the data presented.

Reviewer #1: No

Reviewer #2: Yes

Reviewer #3: Yes

3. Has the statistical analysis been performed appropriately and rigorously?

Reviewer #1: No

Reviewer #2: Yes

Reviewer #3: Yes

4. Have the authors made all data underlying the findings in their manuscript fully available (please refer to the Data Availability Statement at the start of the manuscript PDF file)?

Reviewer #1: Yes

Reviewer #2: Yes

Reviewer #3: Yes

5. Is the manuscript presented in an intelligible fashion and written in standard English?

Reviewer #1: Yes

Reviewer #2: Yes

Reviewer #3: Yes

6. Review Comments to the Author

Reviewer #1: I am disappointed that the authors have elected not to carry out the (not very onerous) additional analysis that I feel is required to permit the conclusions drawn. I cannpot support the pubilcation of this manuscript as is.

Reviewer #2: My comments have been addressed

Reviewer #3: PLOS Global Public Health

PGPH-D-24-01377R1. Re-thinking “non-response” to wasting treatment: exploratory analysis from 14 studies

General reviewer comment:

The research question is interesting and important.

Specific reviewer questions/comments:

1. “Children who were severely underweight on admission had a higher risk of being in the low growth non- response group.”

This statement in the manuscript seems contrary to what is shown on Table 1: at baseline, 78% of High Growth Non-Responders were SAM, while 50% of Low Growth Non-Responders were SAM.

Furthermore, supplemental figures show that baseline anthropometry measures of Low Growth Non-Responders were closer to those of Recovered children. Please clarify.

2. Age of child at baseline is not evenly distributed among the three study groups, as shown on Table 1. High Growth Non-Responder children were significantly more likely to be in the 6-11 month-old group as compared to the other two groups at baseline. Low Growth Non-Responder children were much more likely to be in the 24-59 month-old group. A suggestion is to include discussion of how age differences could have affected the trajectories of growth recovery in the three study groups, and how this could be related to gastrointestinal changes that occur as a reaction to wasting, especially when wasting is prolonged before feeding treatment is started.

Did the authors consider a case-control study, which could have adjusted for age and possibly other factors?

3. It would be important to include discussion of the link between wasting and the status of the gastrointestinal tract (including the possibility of villous blunting, small bowel bacterial overgrowth, exocrine pancreatic insufficiency, among others) and how that affects growth trajectories and response to food-based treatment regimens. Also, how this could be the case especially in the Low Growth Non-Responders who may need further medical assessment and additional non-food related treatment.

4. Given that the journal PGPH is oriented to global public health, it would seem useful to include text in the background section on the overall magnitude of SAM and MAM in children in LMICs, and on the general issue of early prevention of moderate and severe acute malnutrition in children.

5. The manuscript includes some suggestions of ways to follow-up the Low Growth Non-Responders. It could appear not useful to the tone of the manuscript to mention cost-implications of prolonging care of low-growth non-responders which could be interpreted as a deterrent to prolonged care. Discussion of cost could more positively be incorporated as the cost-effectiveness of early prevention, early identification, and early treatment of wasting.

7. PLOS authors have the option to publish the peer review history of their article (what does this mean? ). If published, this will include your full peer review and any attached files.

**Do you want your identity to be public for this peer review?** For information about this choice, including consent withdrawal, please see our Privacy Policy .

Reviewer #1: No

Reviewer #2: **Yes: ** James A Berkley

Reviewer #3: **Yes: ** Laura C Altobelli

---

## [Decision Letter · Decision Letter 2]

13 Nov 2024

PGPH-D-24-01377R2

Re-thinking “non-response” to wasting treatment: exploratory analysis from 14 studies

Dear Dr. James,

Thank you for submitting your manuscript to PLOS Global Public Health. After careful consideration, we feel that it has merit but does not fully meet PLOS Global Public Health’s publication criteria as it currently stands. Therefore, we invite you to submit a revised version of the manuscript that addresses the points raised during the review process.

We look forward to receiving your revised manuscript.

Kind regards,

Dickson Abanimi Amugsi, PhD

Academic Editor

Journal Requirements:

Additional Editor Comments (if provided): 

1. Reviewer #3 notes potential confusion in the manuscript regarding growth categories and their implications on treatment expectations, particularly in the statement about severely underweight children (WAZ <-3). The reviewer highlights that this statement in the abstract sets an expectation for “treatment failures” among children with severe underweight, which may be confusing based on the results presented in Table 1. They recommend an expanded explanation beyond the regression model findings.

2. Additionally, Reviewer #3 suggests that a discussion on age-related differences in outcomes for children starting feeding programs at younger versus older ages, supported by relevant literature, would be valuable.

Reviewers' comments:

Reviewer's Responses to Questions

**Comments to the Author**

1. If the authors have adequately addressed your comments raised in a previous round of review and you feel that this manuscript is now acceptable for publication, you may indicate that here to bypass the “Comments to the Author” section, enter your conflict of interest statement in the “Confidential to Editor” section, and submit your "Accept" recommendation.

Reviewer #3: (No Response)

2. Does this manuscript meet PLOS Global Public Health’s publication criteria ? Is the manuscript technically sound, and do the data support the conclusions? The manuscript must describe methodologically and ethically rigorous research with conclusions that are appropriately drawn based on the data presented.

Reviewer #3: Yes

3. Has the statistical analysis been performed appropriately and rigorously?

Reviewer #3: Yes

4. Have the authors made all data underlying the findings in their manuscript fully available (please refer to the Data Availability Statement at the start of the manuscript PDF file)?

Reviewer #3: Yes

5. Is the manuscript presented in an intelligible fashion and written in standard English?

Reviewer #3: Yes

6. Review Comments to the Author

Reviewer #3: PGPH-D-24-01377R1. Re-thinking “non-response” to wasting treatment: exploratory analysis from 14 studies

Re-Review by Reviewer #3: (see responses in blue)

Reviewer comment1: The research question is interesting and important.

Author response: We are very grateful to reviewer 3 for her comments and insight, which gave us the opportunity to clarify some of the findings and allow us to enhance and supplement the discussion with important additional insights.

Specific reviewer questions/comments:

1. Reviewer comment1: “Children who were severely underweight on admission had a higher risk of being in the low growth non- response group.”

This statement in the manuscript seems contrary to what is shown on Table 1: at baseline, 78% of High Growth Non-Responders were SAM, while 50% of Low Growth Non-Responders were SAM.

Furthermore, supplemental figures show that baseline anthropometry measures of Low Growth Non-Responders were closer to those of Recovered children. Please clarify.

Author response: We thank the reviewer for giving the opportunity to clarify this point.

In the univariate analysis in Table 1, we found that children with MAM (compared to those with SAM) had a lower risk of being in both the high growth non-responder (NR) group (OR 0.69 [95% CI: 0.59, 0.80]) and the low growth NR group (OR 0.19 [0.17, 0.22]) compared to the recovered group. This result is consistent with the descriptive results mentioned: 78% of the high growth NRs were children with SAM, while 50% of the low growth NRs had SAM.

The univariate analysis in Table 1 shows that children who were severely underweight had a higher risk of being in both NR groups compared to being in the recovered group (OR 1.64 for the high growth NRs and OR 1.51 for the low growth NRs). However, when we consider the multivariate analysis (shown in Figure 6), we see the severe underweight criteria only remains significant for predicting those in the low growth NR group. This means that after adjusting for sex, age, MUAC, WHZ at admission, MUAC or weight change at least once during the first month, and infectious morbidity during the treatment, children with WAZ<-3 at admission remain more likely to belong to the low-growth NR group (aOR: 1.6 [1.2-2.0]).

Regarding the baseline anthropometry measures of low growth NRs being closer to those of children who recovered, we found slightly different starting points according to the anthropometric indicator and category of acute malnutrition. For example, in Figure 3 (showing all children), children in the recovered and low growth NR responder groups indeed started with a similar MUAC at baseline (adjusted for age). However, the low growth NR group’s WHZ and WAZ at baseline, after adjusting for age, was slightly lower than for those who recovered. Then in Figure 4 (growth curves among children with SAM only), MUAC at baseline is slightly higher in the low growth NR group compared to the recovered group. In summary, we aimed to describe the shape of the growth curves over time in the narrative, as we felt this was where the interesting differences between the curves, with implications for programming, became apparent.

Reviewer response2: Thank you for this explanation. However, the statement remains confusing as most children in the 3 growth categories were severely underweight (WAZ<-3). This statement in the abstract that I asked about is important since it leads the reader to expect that the children with severe underweight (i.e. WAZ<-3) at baseline will be those most at risk of being “treatment failures” as you call them, and sets up expectations for such to occur for those children. The confusion is that Table 1 shows that, compared to Low Growth NR and Recovereds, High Growth NR has at baseline: highest percentage of WAZ <-3; lowest mean WAZ; lowest weight; highest percentage SAM; and highest % MUAC <115 cm. The multiple regression model indeed shows that LG-NR has significantly higher OR of WAZ <-3 versus the recovered. This should be explained beyond what the regression shows.

2. Reviewer comment1: Age of child at baseline is not evenly distributed among the three study groups, as shown on Table 1. High Growth Non-Responder children were significantly more likely to be in the 6- 11 month-old group as compared to the other two groups at baseline. Low Growth Non- Responder children were much more likely to be in the 24-59 month-old group. A suggestion is to include discussion of how age differences could have affected the trajectories of growth recovery in the three study groups, and how this could be related to gastrointestinal changes that occur as a reaction to wasting, especially when wasting is prolonged before feeding treatment is started.

Did the authors consider a case-control study, which could have adjusted for age and possibly other factors?

Author response: We absolutely agree with the reviewer that age at admission is a major characteristic that can influence the growth curves. For this reason, we included age as an adjustment variable in all models of growth for MUAC and weight. For WHZ, the performance tests on growth curves showed better predictive results without adjustment on age at baseline. We considered whether different age categories influenced the likelihood of being in a certain NR group in figure 6. After adjusting for all other variables in the model, we found that older children (compared to the youngest strata of 6-11 months) had a reduced odds of being in both the low growth NR and high growth NR groups compared to children who recovered. It was interesting that the direction of effect sizes by age did not change between the two groups in figure 6. Hence, we feel we did appropriately consider age in our models.

We did not consider a case-control study because our analyses were already adjusted for age, both for growth curves and the multivariate model. However, the point about gastrointestinal changes is well taken, thank you, and we address this in the next comment.

Reviewer response2: Thank you for this explanation. However, it would be useful to include more discussion on the differences that could be expected between children who begin feeding programs at a younger age versus an older age, and why. No doubt there is literature available that could be cited.

3. Reviewer comment1: It would be important to include discussion of the link between wasting and the status of the gastrointestinal tract (including the possibility of villous blunting, small bowel bacterial overgrowth, exocrine pancreatic insufficiency, among others) and how that affects growth trajectories and response to food-based treatment regimens. Also, how this could be the case especially in the Low Growth Non-Responders who may need further medical assessment and additional non-food related treatment.

Author response: We are grateful to the reviewer for raising this important point. We have amended the discussion section accordingly in lines 372-375. (Note to editorial team, two references have been added accordingly).

Reviewer response2: Thank you for amending your text on this point.

4. Reviewer comment1: Given that the journal PGPH is oriented to global public health, it would seem useful to include text in the background section on the overall magnitude of SAM and MAM in children in LMICs, and on the general issue of early prevention of moderate and severe acute malnutrition in children.

Author response: We agree and have now included this information as recommended in lines 31-35. (Note to the editorial team that this has resulted in 4 new references added).

Reviewer response2: Thank you for adding this information.

5. Reviewer comment1: The manuscript includes some suggestions of ways to follow-up the Low Growth Non- Responders. It could appear not useful to the tone of the manuscript to mention cost-implications of prolonging care of low-growth non-responders which could be interpreted as a deterrent to prolonged care. Discussion of cost could more positively be incorporated as the cost-effectiveness of early prevention, early identification, and early treatment of wasting.

Author response: We thank the reviewer for this suggestion and have included this positive view in the Discussion, lines 438-440.

Reviewer response2: Thank you for making this modification.

7. PLOS authors have the option to publish the peer review history of their article (what does this mean? ). If published, this will include your full peer review and any attached files.

**Do you want your identity to be public for this peer review?** For information about this choice, including consent withdrawal, please see our Privacy Policy .

Reviewer #3: **Yes: ** Laura C Altobelli

---

## [Decision Letter · Decision Letter 3]

30 Dec 2024

Re-thinking “non-response” to wasting treatment: exploratory analysis from 14 studies

PGPH-D-24-01377R3

Dear Dr James,

We are pleased to inform you that your manuscript 'Re-thinking “non-response” to wasting treatment: exploratory analysis from 14 studies' has been provisionally accepted for publication in PLOS Global Public Health.

Best regards,

Dickson Abanimi Amugsi, PhD

Academic Editor

Reviewer Comments (if any, and for reference):

Reviewer's Responses to Questions

**Comments to the Author**

1. If the authors have adequately addressed your comments raised in a previous round of review and you feel that this manuscript is now acceptable for publication, you may indicate that here to bypass the “Comments to the Author” section, enter your conflict of interest statement in the “Confidential to Editor” section, and submit your "Accept" recommendation.

Reviewer #3: All comments have been addressed

2. Does this manuscript meet PLOS Global Public Health’s publication criteria ? Is the manuscript technically sound, and do the data support the conclusions? The manuscript must describe methodologically and ethically rigorous research with conclusions that are appropriately drawn based on the data presented.

Reviewer #3: Yes

3. Has the statistical analysis been performed appropriately and rigorously?

Reviewer #3: Yes

4. Have the authors made all data underlying the findings in their manuscript fully available (please refer to the Data Availability Statement at the start of the manuscript PDF file)?

Reviewer #3: Yes

5. Is the manuscript presented in an intelligible fashion and written in standard English?

Reviewer #3: Yes

6. Review Comments to the Author

Reviewer #3: (No Response)

7. PLOS authors have the option to publish the peer review history of their article (what does this mean? ). If published, this will include your full peer review and any attached files.

**Do you want your identity to be public for this peer review?** For information about this choice, including consent withdrawal, please see our Privacy Policy .

Reviewer #3: **Yes: ** Laura C Altobelli, DrPH, MPH
